# Unravelling the Antifibrinolytic Mechanism of Action of the 1,2,3-Triazole Derivatives

**DOI:** 10.3390/ijms25137002

**Published:** 2024-06-26

**Authors:** Yvette Rabadà, Oriol Bosch-Sanz, Xevi Biarnés, Javier Pedreño, Luis Caveda, David Sánchez-García, Jordi Martorell, Mercedes Balcells

**Affiliations:** 1IQS School of Engineering, Universitat Ramon Llull, Via Augusta 390, 08017 Barcelona, Spain; yvetterabadas@iqs.url.edu (Y.R.); oriol.bosch@iqs.url.edu (O.B.-S.); david.sanchez@iqs.url.edu (D.S.-G.); jordi.martorell@iqs.url.edu (J.M.); 2Institute of Medical Engineering and Science, Massachusetts Institute of Technology, 77 Massachusetts Ave., Cambridge, MA 02139, USA; javierpedreno@hotmail.com; 3Laboratory of Biochemistry, Institut Químic de Sarrià, Universitat Ramon Llull, Via Augusta 390, 08017 Barcelona, Spain; xavier.biarnes@iqs.url.edu; 4Alxerion Biotech, 245 First St, Riverview II, 18th Floor, Cambridge, MA 02142, USA; luiscaveda@gmail.com; 5Grup d’Enginyeria de Materials, Institut Químic de Sarrià, Universitat Ramon Llull, Via Augusta 390, 08017 Barcelona, Spain

**Keywords:** fibrinolysis, tranexamic acid, lysine analogue, antifibrinolytic agents, triazole, plasmin, plasminogen

## Abstract

A new family of antifibrinolytic drugs has been recently discovered, combining a triazole moiety, an oxadiazolone, and a terminal amine. Two of the molecules of this family have shown activity that is greater than or similar to that of tranexamic acid (TXA), the current antifibrinolytic gold standard, which has been associated with several side effects and whose use is limited in patients with renal impairment. The aim of this work was to thoroughly examine the mechanism of action of the two ideal candidates of the 1,2,3-triazole family and compare them with TXA, to identify an antifibrinolytic alternative active at lower dosages. Specifically, the antifibrinolytic activity of the two compounds (**1** and **5**) and TXA was assessed in fibrinolytic isolated systems and in whole blood. Results revealed that despite having an activity pathway comparable to that of TXA, both compounds showed greater activity in blood. These differences could be attributed to a more stable ligand–target binding to the pocket of plasminogen for compounds **1** and **5**, as suggested by molecular dynamic simulations. This work presents further evidence of the antifibrinolytic activity of the two best candidates of the 1,2,3-triazole family and paves the way for incorporating these molecules as new antifibrinolytic therapies.

## 1. Introduction

Fibrinolysis is a physiological process that enables restoration of the vascular system’s permeability. Following vascular damage, a fibrin clot is formed at the site of injury and, once bleeding is stopped and wound repair has occurred, the clot is dissolved through the fibrinolytic system [1,2,3]. This mechanism counteracts coagulation and is essential to maintain the hemostatic balance, avoiding excessive fibrin formation and obstruction of blood vessels [4,5,6,7]. Plasmin is the enzyme accountable for fibrin cleavage and circulates in plasma as the zymogen plasminogen. Plasminogen is converted into its activated form by tissue-type plasminogen activator (tPA) and, to a lesser extent, by urokinase-type plasminogen activator (uPA) [8]. However, circulating plasminogen displays a closed conformation with little activation capacity in the absence of fibrin. The lysine binding sites (LBSs) in plasminogen’s Kringle domains mediate the binding to fibrin or lysine analogues [8,9]. Hence, tPA and plasminogen co-localize on the surface of a fibrin clot, forming a “ternary complex” [10,11,12]. This binding induces a conformational change in the structure of plasminogen, adopting an unfolded form that can be converted into plasmin [8,12,13,14]. After the two serine proteases bind to fibrin, the catalytic efficiency of tPA is enhanced, and it activates plasminogen significantly, enabling a localized breakdown of fibrin to fibrin degradation products (FDPs) [14,15].

Dysregulation of the fibrinolytic system is related to numerous life-threatening conditions, either due to an impaired or an excessive activity of the system [1]. Hyperfibrinolysis is a pathological state associated with greater mortality and morbidity and constitutes the etiology of many hemorrhagic conditions [14,16]. Epistaxis, mucosal bleeding in patients with coagulopathies, menometrorrhagia, severe postpartum hemorrhage, and postoperative bleeding after surgery are different examples of hyperfibrinolytic states resulting in mild to moderate bleeding [16,17,18]. Systemic hyperfibrinolysis is also triggered in traumatic coagulopathy and subsequent bleeding is the first cause of preventable death among injured patients [19].

Antifibrinolytic agents such as tranexamic acid (TXA) or ε-aminocaproic acid (EACA) have proven effective at reducing hemorrhagic diatheses [5,14,20]. TXA, in particular, decreases perioperative blood loss and transfusion requirements in major surgeries [16,21]. In addition, early administration of TXA increases survival rates in traumatic hemorrhage, especially in patients with hemorrhagic shock [22]. These two synthetic lysine analogues are members of the serpin superfamily [23]. They intercept the natural sequence of fibrinolysis by reversibly binding to the LBS of plasminogen, displacing the binding of plasminogen to fibrin and, by that, preventing its conversion to plasmin [23,24,25,26,27]. In addition to being an inhibitor of the LBS of plasminogen, TXA is also reported to directly inhibit plasmin, even though affinity for its catalytic domain is only described at greater concentrations [28].

Although TXA is considered an effective antifibrinolytic, and is generally well tolerated at the usual dosage, some studies present doubts regarding its safety profile [29,30,31]. Concerns have increased over the years about TXA promoting a hypercoagulable state, associated with an increased risk of thrombotic complications. Previous analyses concluded that TXA might intensify the risk of venous thromboembolism [32,33], even though the overall thrombotic capacity still remains unclear [34,35]. TXA is also related to a 7-fold increased incidence of postoperative seizures, mostly in cardiac surgery patients, but an extensive range of patients exposed to TXA may be susceptible [21,29,36,37,38]. Such postoperative seizures have been associated with a 2.5-fold increase in the intrahospital mortality rate [35,39,40]. TXA-associated seizures result from the drug’s capacity to cross the blood–brain barrier, resulting in cerebrospinal fluid concentrations that are approximately 10% of TXA’s plasma concentration [28,29,35]. Dosage is, hence, one of the most critical risk factors in TXA-associated seizures. Patients with renal dysfunction are also at greater risk due to a slower clearance of TXA, increasing the drug’s cerebrospinal fluid concentration. TXA dosage in these patients must be adjusted to serum creatinine levels, when not directly contraindicated [37,39,40,41,42]. This, combined with the lack of reliable antifibrinolytic alternatives in the market, highlights the need to develop new options active at lower doses [35,36].

A recent work by Bosch-Sanz et al. studied a new family of fibrinolysis inhibitors, with the goal to overcome current limitations. A set of compounds combining a piperidine ring, a triazole, and an oxadiazolone ring were identified as potential antifibrinolytic drugs [43]. In comparison to TXA, these molecules contained two more rigid rings in their structure, with the aim of increasing their antifibrinolytic activity and selectivity. Docking simulations between these compounds and the LBS of plasminogen reported that the terminal piperidine of these compounds interacted with the acidic side of the pocket of plasminogen’s LBS, while the modified oxadiazolone ring interacted with the basic side [43]. These target–ligand interactions are similar to those of the traditional lysine analogues, but the presence of a triazole linker provided additional interactions which enhanced the binding affinity to the LBS. However, the 1,2,3-triazole moiety proved to be crucial for the antifibrinolytic activity of molecules **1** and **5** in comparison to its 1,2,4-triazole counterpart at concentration ranges similar if not lower than TXA’s. Molecular dynamics analysis revealed that the 1,2,3-triazole moiety provided anchoring to the binding site, which the 1,2,4-triazole ring could not provide, resulting in a higher stabilizing capacity of the 1,2,3-triazole ring for the LBS [43]. Therefore, given the correlation between the 1,2,3-triazole derivatives and TXA regarding their interactions with the LBS of plasminogen, it is possible that this new family of compounds exhibits a similar behavior towards the components of the fibrinolytic system. Thus, we hypothesize that the mechanism of action of these novel 1,2,3-triazole derivatives is analogous to that of TXA, the current gold standard. However, we expect a higher target specificity, attributed to an enhanced ligand–target binding affinity.

In the present work, we investigated the mechanism of action of compounds **1** and **5**, the two 1,2,3-triazole derivatives that contain an oxadiazole and a piperidine ring (Figure 1), and compared them to TXA through a series of assays. For this purpose, isolated enzyme assays and ex vivo assays with whole blood were performed to determine the effects of the tested compounds on the fibrinolytic cascade. Finally, an in vivo pharmacokinetic proof-of-concept study was conducted to assess the profile and safety of compound **5**. By combining these techniques, we aimed to provide a thorough analysis of the mechanism of action of compounds **1** and **5** and provide insights into their potential application as antifibrinolytic agents.

## 2. Results

### 2.1. Isolated Enzyme Assays

The ability of compounds **1** and **5** to inhibit plasmin’s cleavage of a synthetic substrate was assessed and compared to TXA’s. As seen in Table 1, compounds **1**, **5**, and TXA displayed an inhibitory effect on plasmin’s active site. The IC_50_, representing the concentration at which each compound reduced plasmin’s activity by half, was significantly lower (*p* < 0.05) for **1** and **5** in comparison to TXA, but in a similar order of magnitude. Compounds **1** and **5**, as well as TXA, inhibited the catalytic domain of plasmin in a dose-dependent manner, showing that plasmin’s active site was clearly susceptible to increasing concentrations of the tested molecules (Appendix A). Next, the ability of both compounds to inhibit tPA’s activity was assessed and compared to that of TXA’s (Appendix A). Neither TXA nor the compounds were able to inhibit tPA, and thus block the active site of tPA. At therapeutic doses, compounds **1** and **5** showed no clear activity on the catalytic domain of tPA.

As described in the literature, TXA promotes the conversion of plasminogen into plasmin by tPA in the absence of fibrin [44,45]. This capacity was also studied in compounds **1** and **5** and compared to TXA’s in a plasminogen-tPA isolated system. Figure 2 shows that the activation rate was dose-dependent for the three molecules, and **1** and **5** were able to accelerate the conversion of plasminogen into plasmin at significantly lower concentrations than TXA. Indeed, the concentration at which each compound enhanced tPA’s activity by 50% demonstrated that **1** and **5** are more than 50 times more potent than TXA in catalyzing plasminogen’s activation by tPA.

### 2.2. Ex Vivo Assays

An ex vivo whole blood clotting test was performed to determine whether compounds **1** and **5** had any influence on clot formation. Blood was allowed to clot in tubes containing the studied compounds or TXA, and was compared to a control clot. A visual representation of clot formation is shown in Figure 3. The study was performed without the addition of exogenous tPA and, therefore, fibrinolysis was not triggered. When blood was added to the tubes, blood clots were generated at the same time for the four different conditions. After 10 min of incubation at 37 °C, clots were completely formed and contained all the blood present in the tubes. Visual differences in clot formation amongst the four different conditions were not observed, suggesting that coagulation was not susceptible to the activity of compounds **1**, **5**, or TXA. After 30 min of incubation, fibrinolysis of the four clots was negligible, reflecting that the amount of endogenous tPA in blood was insufficient to stimulate spontaneous clot lysis. Therefore, evidence suggested that products **1** and **5**, as well as TXA, do not have any effect on the mechanism of clot formation when fibrinolysis is not triggered.

The antifibrinolytic activity of compounds **1** and **5** was then assessed in whole blood and compared to that of TXA’s, to study their effect when fibrinolysis was triggered after the addition of exogenous tPA. Firstly, the effective dosage of compounds **1**, **5**, and TXA was determined in whole blood and compared to an antifibrinolytic-free control clot. The effective dosage was defined as the concentration at which each molecule prevented the complete degradation of the clot after 24 h of incubation. A visual portrayal of clot lysis incubated with different concentrations of the tested compounds is shown in Figure 4. Control clots were fully lysed after 24 h of incubation, implying that fibrinolysis activators triggered the fibrinolytic system once the clot was formed. However, when TXA or compounds **1** and **5** were added to the tubes, clots were still visible after 24 h, manifesting the antifibrinolytic activity of the three compounds. Among the tested molecules, products **1** and **5** had a prominent antifibrinolytic activity, which was highly evident in the whole range of concentrations. TXA was only active after 24 h at 60 µM, although the remaining clot size was smaller than that of compounds **1** and **5**.

The antifibrinolytic activity of compounds **1** and **5** was also studied on whole blood by measuring the concentration of D-dimer, a specific fibrinolytic marker released in clot lysis [17]. After the addition of tPA and the two compounds or TXA on pre-formed clots, the concentration of released D-dimer was quantified and compared to a non-treated control clot (Figure 5). D-dimer levels increased in the four different conditions after tPA was added to the clots, implying that fibrinolysis had been initiated. After 24 h of incubation, a significant increase in D-dimer was detected in the control clot compared to clots containing one of the studied compounds or TXA, evidencing that the two studied compounds, as well as TXA, were effective inhibitors of clot lysis. D-dimer release at timepoint 24 h was significantly lower for compound **1** than for TXA and around 35% lower than for compound **5**, and was the molecule with more antifibrinolytic potency according to this test.

### 2.3. Computational Analysis

Molecular dynamics simulations were performed for the three studied compounds to compare the behavior of **1** and **5** to TXA against the lysine binding site in the Kringle 1 domain of plasminogen (Figure 6). Kringle domains are peptide regions that mediate plasminogen’s interactions with other entities, with Kringle 1 being the most significant.

Molecular dynamics results showed different behaviors between TXA and compounds **1** and **5** (Figure 6). Among the three tested compounds, TXA was the first molecule to dissociate from the lysine binding site. It left the pocket before the simulation time of 150 ns, which was revealed by the substantial increase in distance between the terminal amine of TXA and Asp57. In the case of compounds **1** and **5**, the equivalent distance between the piperidine amine and Asp57 remained between 3 and 7 Å throughout practically the entire simulation. This distance range is indicative of polar interactions, and at the lower range it also points to the formation of H-bonds. Additionally, the nitrogen atoms from the triazole rings of molecules **1** and **5** provided an anchoring effect due to their interaction with the polar head group of residue Tyr72. For both compounds, such distances persisted between 6 and 10 Å throughout the entire simulation, allowing for transient polar interactions. Such additional anchoring was not present in the case of TXA, due to the lack of functional groups in TXA that could mediate interactions with Tyr72.

### 2.4. In Vivo Assays

The pharmacokinetic behavior of compound **5** was studied in Wistar rats and in Beagle dogs. Figure 7 displays the concentration profile of compound **5** in plasma at various dose levels for both male and female rat and dog models.

No mortality or morbidity was observed in any of the dosing groups for rats, and they did not show any adverse clinical sign during the 24 h study period (Appendix A). According to Figure 7, male and female rats of each respective group showed a similar decrease in the concentration of compound **5** in plasma over time, displaying similar concentration values at each timepoint. At the final timepoint of 24 h, female rats of all dose levels did not present any detectable concentration and only one of the male rats of each dose level still showed traceable concentrations (Appendix A). Additionally, different pharmacokinetic parameters were determined for each group. Maximum concentration (C_max_) of the tested compound in rats was observed 5 min after intravenous administration at all the different dose levels (Appendix A). C_max_ was higher in the three female dose groups when compared to that of male rats. However, elimination halftime (t_1/2_) was much higher in male rats in comparison to female rats, showing different elimination rates for both sexes. Data showed that there was no saturation at the highest dose tested for both sexes since C_max_ and area under the curve (AUC) increased accordingly with the dose administered.

In the case of Beagle dogs, a single dose level of 5 mg/kg body weight (b. wt.) was studied for each sex. No mortality or morbidity was observed amongst study dogs (Appendix A). Dogs did not show any adverse clinical sign during this period, except for one male and one female dog, which presented mild vomiting immediately after dosing. Results displayed in Figure 7 also reveal similar pharmacokinetic profiles for both sexes, with compound **5** still being detected after 24 h. C_max_ was achieved 5 min after administration, with females also presenting higher values (Appendix A). Data did not show substantial differences in the elimination halftime between male and female dogs.

After studying the pharmacokinetic profile of compound **5**, the maximum tolerated dose (MTD) was also determined in Wistar rats and in Beagle dogs to evaluate the potential toxicity of compound **5** after a single administration. In rats, a single dose study with two dose levels, 300 (D1) and 500 (D2) mg/kg b. wt., was conducted. After a single intravenous injection, rats were closely monitored for a period of 14 days to detect signs of toxicity. No mortality or morbidity was observed for either of the dose levels throughout the study period (Appendix A). Additionally, weight loss was not detected in any of the groups since mean body weight increased accordingly in rats treated with both doses (Appendix A). Six of the ten rats at the D1 dose level developed erythema at the injection site (Appendix A). However, all D2 rats, which received the highest dose of compound **5**, presented tail erythema, which was followed by tail necrosis and sloughing in 7/10 rats, with a higher incidence in female rats. Regarding macroscopic findings observed after external gross examination of sacrificed rats, D1 male and female rats did not reveal any abnormality (Appendix A). In the case of D2 rats, gross examination exposed reddish discoloration and sloughing of the tail in 2/5 male and 5/5 female rats. This higher incidence of pathological findings observed in female rats was consistent with the increased frequency of clinical signs shown by female rats. Internal gross examination of male and female rats from D1 and D2 did not reveal any abnormality.

In Beagle dogs, MTD was also determined through an intravenous bolus injection of compound **5**. Two different dose levels were tested in this case: 50 mg/kg (D3) and 75 mg/kg (D4) b. wt. Toxicity signs and clinical parameters were determined for a period of 7 days after dosing. During this period, no mortality nor morbidity was observed (Appendix A). D3 dogs presented vomiting after dosage on day 1, which was relieved the same day (Appendix A). In a similar way, vomiting and weakness were observed in D4 dogs after dosing. While vomiting remitted during the same day, weakness disappeared one day after. No differences were observed between sex regarding the clinical sings for each group in dogs. In addition, body weight increased as per usual and changes in food consumption related to treatment were not detected for any of the dose groups for 7 days (Appendix A). Dogs of both groups did not show any alteration related to the tested molecule in hematology and clinical chemistry parameters on day 7 after dosing (data available on demand). Clinical pathology analysis at terminal sacrifice was also performed on dogs and no treatment-related alterations were observed in terminal body and organ weight (data available on demand). Concerning the pathological findings on D4 dogs after sacrifice, external and internal macroscopic examination did not reveal any abnormality (Appendix A). However, microscopic examinations showed that D4 male dogs presented alterations in the liver, kidneys and testes, while D4 female dogs only showed abnormalities in the liver.

## 3. Discussion

Bosch-Sanz et al. identified a new family of antifibrinolytic drugs containing a 1,2,3-triazole ring and showed their antifibrinolytic properties in plasma [43]. Evidence suggested that this family of compounds acted, like TXA, in the lysine binding sites of plasminogen, blocking its activation as plasmin and the subsequent fibrinolysis. However, effects on other fibrinolytic components could not be discounted. The present study demonstrates the antifibrinolytic mechanism of action of compounds **1** and **5**, two analogous 1,2,3-triazole derivatives, which showed increased antifibrinolytic activity in plasma in comparison to their counterparts. These two triazole derivatives contain an oxadiazolone and a piperidine ring.

The activity of compounds **1** and **5** was tested in the presence of different fibrinolytic elements to identify their enzymatic target. Both compounds proved to be inhibitors of the active site of plasmin in the same order of magnitude as TXA. These values were consistent with previous findings reporting that TXA is a noncompetitive inhibitor of plasmin at high doses [14,25,27]. Aprotinin is a serine protease that strongly inhibits plasmin, but this modulation is achieved at much lower concentrations [46,47]. Therefore, compounds **1** and **5**, as well as TXA, have less affinity than the natural inhibitor aprotinin for plasmin. In addition, compounds **1**, **5**, and TXA showed a 1000-fold increase in the values of the IC_50_ of plasmin’s catalytic domain compared to the IC_50_ values obtained in the plasma clot lysis assay by Bosch-Sanz et al. [43], showing that the inhibitory effect on plasmin takes place at nontherapeutic doses. Further studies were performed to analyze the effect of both molecules on the plasminogen activator tPA and compare it to TXA’s, which has been reported to have a negligible inhibitory effect at non-clinical concentrations [27]. Accordingly, results showed that compounds **1**, **5,** and TXA have no activity on tPA’s active site at the tested concentrations. This dismisses the hypothesis that this serine protease is the clinical target enzyme for the triazole derivatives. In line with other studies performed for lysine analogues in the absence of fibrin [44,45], compounds **1** and **5** stimulated the conversion of plasminogen into plasmin by tPA in a dose-dependent manner. These results strengthened the hypothesis that molecules **1** and **5** share the same mechanism of action as the lysine analogue TXA. In this artificial state, the concentration at which compounds **1** and **5** enhanced tPA’s activity by 50% provided a more than 50-fold improvement over TXA, showing higher affinity for plasminogen than the current gold standard.

The possibility of the 1,2,3-triazole derivatives having an effect on clot formation was also explored. Both drugs were assessed on clot formation in the absence of a plasminogen activator, and, in consequence, without triggering fibrinolysis. Tranexamic acid is not reported to have any effect on the in vitro coagulation time, since it solely acts on the fibrinolytic pathway at therapeutic doses [48]. Indeed, differences were not observed in the formation of blood clots, either for compounds **1** and **5** or TXA at a concentration of 40 µM, suggesting that none of the drugs affect the clotting phase. These findings provided qualitative evidence that compounds **1** and **5** exert their effect exclusively on the fibrinolytic pathway. Further whole blood assays, in which fibrinolysis was triggered, allowed the assessment of the antifibrinolytic activity of the studied molecules taking into consideration all circulating cells, which play a major role in thrombolysis and in clot properties and structure [49,50]. Results from these assays reinforced the antifibrinolytic capacity of compounds **1** and **5** and showed higher activity in comparison with TXA at lower doses in whole blood. **1** and **5** were capable of preventing clot lysis at the concentration of 10 µM in our assay, while the effective dosage for TXA was 60 µM. In addition, D-dimer quantification revealed an improvement of more than 2-fold in the antifibrinolytic activity of compound **1** over TXA, which is a very similar result to that obtained in plasma by Bosch-Sanz et al. [43]. Even though compound **5** demonstrated higher antifibrinolytic activity than TXA when assayed in whole blood, this was not consistent with previous results obtained in plasma [43]. Therefore, the greater activity of compound **1** over TXA was demonstrated in plasma and in whole blood, but compound **5** only showed higher antifibrinolytic potency in blood. Further experimental investigations are needed to clarify differences in the antifibrinolytic activity on plasma and whole blood, and assess the possible effect of the blood components over the studied molecules. Molecular dynamics simulations evidenced the heterogeneous behavior of molecules **1**, **5**, and TXA in the lysine binding site of plasminogen. These results manifested that the triazole derivatives had a more stable ligand–target binding in comparison to TXA, which left the pocket during simulation time. The presence of the nitrogen atoms in the triazole rings and the secondary amine at the piperidine ring might be accountable for the longer stabilizing capacity for the pocket and therefore, for the higher antifibrinolytic activity.

As an in vivo proof of concept, pharmacokinetic and maximum tolerated dose (MTD) studies were performed in Wistar rats and Beagle dogs for compound **5**. This compound was selected based on its patentability, as it had not been described before, and on the fact that its synthetic route was completed in less steps and with a much better overall yield. The pharmacokinetic study in the rat and dog models showed similar results, providing an elimination halftime between 2 and 3 h for most of the concentrations, in the same order of magnitude as reported for EACA and TXA [51]. The MTD assay did not cause any mortality or morbidity in rats or dogs. The final MTD was determined based on non-lethal symptoms, such as tail sloughing—in rats—and vomiting and weakness—in dogs. The MTD was hence determined to be 300 mg/kg b. wt. when administered via intravenous injection for Wistar rats. This was the dose level tolerated by both male and female rats, under the conditions and procedures followed in this study. Histopathological findings revealed the presence of cytoplasmic rarefaction in livers of male and female dogs, which could be related to dosing of molecule **5**, but could also be considered adaptive, since related parameters such as organ weight and biochemistry remained normal. Moreover, minimal lesions detected in kidneys of male dog were nonspecific and could be considered to be of spontaneous nature [52]. In this assay, the MTD was determined to be 75 mg/kg b. wt. for both male and female Beagle dogs, when administered intravenously with a bolus injection.

## 4. Materials and Methods

### 4.1. Materials

TXA (Sigma Aldrich, St. Louis, MO, USA) was used without further purification. Compounds **1** and **5** were synthesized in-house as previously described [43]. Tris hydrochloride, 1M, pH = 7.5 solution (Thermo Fisher Scientific, Waltham, MA, USA) was used as buffer. For the in vitro and ex vivo assays, compounds **1**, **5**, and TXA were diluted with Tris HCl buffer from a ddH_2_O stock. Recombinant human Tissue Plasminogen Activator (abcam, Cambridge, UK) served as the source of tPA, and was stored at −20 °C until used.

### 4.2. Isolated Enzyme Assays

#### 4.2.1. Plasmin Activity Assay

The potential activity of the studied compounds on the catalytic domain of plasmin was evaluated using the Plasmin Inhibitor Screening Assay Kit (abcam), following the manufacturer’s instructions. Briefly, the assay measures the inhibition of plasmin’s activity by the tested compounds through the cleavage of a fluorogenic peptide substrate. The assay was performed in flat-bottom 96-well plates. Plasmin was pre-incubated with different concentrations of the tested compounds for 10 min at RT prior to the addition of the peptide substrate. Plasmin’s activity was then measured by monitoring the increase in fluorescence intensity over time using a microplate reader (Infinite M Plex, Tecan, Männedorf, Switzerland) at an Ex/Em of 360/450 nm, every minute at 37 °C for 10 min. The slope of each sample was determined using the initial and final fluorescence values and their respective time points.
(1)$$Fluorescence\;slope=\frac{{RFU}_{2}-{RFU}_{1}}{t_{2}-t_{1}}$$

The Relative Plasmin Inhibition was calculated according to the manufacturer’s instructions using a control slope with no compound (enzyme control):(2)$$Relative\;Plasmin\;Inhibition\left(\%\right)=\frac{{Slope}_{Control}-{Slope}_{Compound}}{{Slope}_{Control}}\times 100$$

IC_50_ values were determined as the concentration at which each molecule was capable of inhibiting plasmin’s activity by 50% of the control.

#### 4.2.2. tPA Activity Assay

The activity of tPA was evaluated in the presence of compounds **1**, **5**, and TXA using the Tissue Plasminogen Activator Activity Assay Kit (BioVision, Milpitas, CA, USA), according to the manufacturer’s instructions. The assay measures the inhibition of tPA’s catalytic domain by the tested compounds employing a chromogenic substrate. Different concentrations of the studied molecules were incubated with tPA in a 96-well plate at 37 °C for 10 min. The final concentration of tPA in each well was 3.25 μg/mL, considering the total volume of 100 μL. After, 20 μL of tPA’s substrate was added to each well. The activity of tPA was assessed by measuring the absorbance on a microplate reader at a wavelength of 405 nm every minute at 37 °C for 30 min. The slope for each assay was obtained from the initial and final absorbance values and their respective time points, or until complete substrate cleavage was achieved.
(3)$$Absorbance\;slope=\frac{{Abs}_{2}-{Abs}_{1}}{t_{2}-t_{1}}$$

The Relative tPA Inhibition for each condition was calculated using Equation (2). Then, the Relative tPA Activity (%) was calculated by subtracting the relative tPA inhibition of each concentration from the total activity of the enzyme, obtained from a control without a compound.

#### 4.2.3. Plasminogen Activation Assay

The conversion rate of plasminogen to plasmin was assessed using the Tissue-Type Plasminogen Activator Activity Assay Kit (abcam), following the manufacturer’s instructions. The assay measures the ability of tPA to activate plasminogen to plasmin in the presence of the tested compounds using a chromogenic substrate.

A solution of plasminogen and plasmin substrate was mixed with a previously prepared solution of the studied compound and tPA in Tris HCl buffer in a 96-well plate. The final concentration of tPA in each well was 6.8 µg/mL, in a final volume of 100 µL. Absorbance was measured on a microplate reader at a wavelength of 405 nm every minute at 37 °C for 2 h or until the system reached saturation. The absorbance slope of each sample was determined using Equation (3). The Relative Plasminogen Activation (%) for each condition was calculated with Equation (4), using a control without a compound (enzyme control).
(4)$$Relative\;Plasminogen\;Activation\;(\%)=\frac{{Slope}_{Compound}-{Slope}_{Control}}{{Slope}_{Control}}\times 100$$

Values were fitted to a linear model using MATLAB R2023a (The Math Works Inc., Natick, MA, USA). Concentrations at which each compound was able to increase plasminogen’s conversion by 50% in comparison to the enzyme control were extracted and compared for the three molecules.

### 4.3. Ex Vivo Assays

#### 4.3.1. Blood Extraction

Blood was obtained by venipuncture from healthy donors after they signed an informed consent form. Blood was transferred into vacuum non-anticoagulant Vacutainer^®^ tubes (BD, Franklin Lakes, NJ, USA). Blood samples were immediately used for the assay after extraction.

#### 4.3.2. Whole Blood Coagulation Assay

The effect in clot formation was studied for compounds **1**, **5**, and TXA. Prior to blood addition, 20 µL of the evaluated compounds was added to glass tubes, in a final concentration of 40 µM. As a control clot, 20 µL of Tris HCl buffer with no compound was used. Next, 800 µL of blood was transferred to each tube followed by casual shaking to ensure proper mixing. Tubes were incubated at 37 °C for 30 min to allow clot formation and posterior lysis. Blood clotting and lysis were visually monitored and photographs were taken after 30 min.

#### 4.3.3. Whole Blood Dosage

The effective dosage of the antifibrinolytic activity of compounds **1**, **5**, and TXA was also studied. Glass tubes were prepared with 20 µL of a mixture of tPA, in a final concentration of 10 µg/mL, and the evaluated compounds, in a final concentration of 10, 20, 40, or 60 µM in Tris HCl buffer. A control clot was obtained with 20 µL of tPA with no compound. Then, 800 µL of blood was transferred to each tube and tubes were monitored for clot formation and lysis. The assay was performed with casual shaking. Tubes were incubated at 37 °C and clots were observed for 24 h or until they were completely lysed. The ex vivo effective dosage for each compound was determined by their ability to inhibit complete clot degradation. Photographs were taken after 24 h of incubation.

#### 4.3.4. Whole Blood Clot Lysis Assay

The quantification of D-dimer released during clot lysis was performed using the human D-dimer ELISA kit (Thermo Fisher Scientific). Directly after extraction, 500 µL of blood was transferred to four different glass tubes and allowed to clot for 10 min at 37 °C. Tubes were incubated at 37 °C for an additional 1 h to enable clot retraction from the walls of the glass tubes [53]. Then, 100 µL of a mixture of tPA and the studied compounds was added on top of each clot. Each tube had a tPA concentration of 10 µg/mL in the final volume and the tested compounds were assessed at a final concentration of 20 µM. As a control clot, 100 µL of tPA without compound in Tris HCl buffer was added. Tubes were incubated at 37 °C and, at time points of 5 min, 2, 4, 7, and 24 h, samples of each tube were extracted for D-dimer quantification [54]. After gentle shaking of the tubes, 5 µL of the supernatant from each tube was pipetted into Eppendorf tubes and stored at −20 °C until assayed. The ELISA assay was performed as described by the manufacturer’s instructions and absorbance was read in a microplate reader at the wavelengths of 450 nm and 550 nm. Values obtained from the higher wavelength were subtracted from the values from the lower wavelength to correct optical imperfections of the microplate. Sample concentrations were obtained from the standard curve. Values were fitted to a nonlinear model using MATLAB software R2023a.

### 4.4. Molecular Dynamics Simulations

Molecular dynamics simulations were performed as previously described [43] with a simulation time of 200 ns for each compound. The protein (PDB code 1cea) was inserted into a cubic box of water molecules, ensuring that the solvent shell would extend for at least 0.8 nm around the system. Three sodium counterions were added. The GROMOS 54a7 force field was used for both the protein and ligands. The water molecules were described by the SPC/E model. Parameters for the ligands were generated with ATB webserver [55]. The system was minimized by imposing harmonic position restraints of 1000 kJ·mol^−1^·nm^−2^ on solute atoms, allowing the equilibration of the solvent without distorting the solute structure. After an energy minimization of the solvent and the solute without harmonic restraints, the temperature was gradually increased from 0 to 298 K. This was performed by increasing the temperature from 0 to 298 K in 12 steps, in which the temperature was increased by 25 K in 100 ps of MD.

Constant temperature−pressure (T = 298 K, P = 1 bar) 200 ns dynamics was then performed through the Nosé–Hoover and Andersen–Parrinello–Rahman coupling schemes. Periodic boundary conditions were applied. The final simulation box was equilibrated at around 5.61 × 5.61 × 5.61 nm. Long-range electrostatic interactions were treated with the particle mesh Ewald (PME) method, using a grid with a spacing of 0.12 nm combined with a fourth-order B-spline interpolation to compute the potential and forces between grid points. The cutoff radius for the Lennard-Jones interactions, as well as for the real part of PME calculations, was set to 0.9. Analysis of the simulated trajectories was performed with VMD.

### 4.5. In Vivo Assays

All tests were performed at Jai Research Foundation (JRF), India. Studies were undertaken in compliance with the guidelines of the Association for Assessment and Accreditation of Laboratory Animal Care (AAALAC), USA.

#### 4.5.1. Pharmacokinetic Study

A pharmacokinetic study was performed for compound **5** in Wistar rats and Beagle dogs. For rats, a total of 35 male and 35 female rats were used for the study. At initiation of the dosing, rats were 9 to 10 weeks old and the body weight variation among rats was within ±20% of the mean body weight for each sex. Rats were acclimated for a period of 5 days prior to randomization and were housed in groups of 2 to 3 rats/cage/sex in an animal room maintained at 21 ± 2 °C and 65 ± 1% relative humidity, on a 12–12 h light–dark cycle (beginning at 06:00 h) with 17 air changes per hour. Food and filtered water were provided ad libitum. During randomization, 30 male and 30 female rats were selected for the study and randomly allocated to the 4 different groups. The remaining 5 male and 5 female rats were returned to the animal facility. Rats were observed twice daily for morbidity, mortality, and clinical signs during the whole study period. Rats were euthanized after last blood collection time point by carbon dioxide asphyxiation. Sterile water served as a vehicle to dissolve compound **5** and the resulting formulations were administered right after preparation. Dose formulations were administered to rats via an intravenous bolus injection through the tail vein, using a sterile needle attached to a graduated syringe. A fixed volume of 5 mL/kg body weight was administered. Individual doses were adjusted based on the most recent recorded body weight of each rat. Four dose levels were selected: vehicle control (PK1), 2.5 mg/kg (PK2), 5 mg/kg (PK3), and 10 mg/kg (PK4).

Pharmacokinetic assessment of compound **5** was conducted on the day of dosing at multiple time points: 0 h (pre-dose), 5 min, 15 min, 0.5 h, 1 h, 2 h, 4 h, 6 h, 10 h, and 24 h post-dosing for PK2, PK3, and PK4 rats. At each time point, samples were only obtained from a subset of 3 rats. For the PK1 group, samples were collected at 0 h and at 24 h post-dosing. Blood samples (400 µL) were collected into heparin sodium tubes from the retro-orbital plexus under very light isoflurane anesthesia. Samples were inverted 4 to 5 times and placed on ice and then centrifuged at 3500 rpm for 15 min at 8 °C for plasma extraction. Plasma was transferred to Eppendorf tubes and stored at −70 ± 10 °C until assayed. The concentration of compound **5** was analyzed in plasma using High-Performance Liquid Chromatography. A Waters-Fluoro-Phenyl column was used (150 × 4.6 mm, 3.5 µm) with a flow rate of 0.6 mL/min. The mobile phase consisted of acetonitrile (A) and 2 mM ammonium Formate in Mili-Q water (B) with an isocratic elution mode. Pharmacokinetic analysis of the concentration of compound **5** in plasma was performed using the non-compartmental model of the WinNonlin^®^ software version 8.1 at JRF. The group mean values of plasma concentration were analyzed to obtain different pharmacokinetics parameters, including C_max_ (maximum peak concentration), T_max_ (time of maximum peak concentration), AUC (area under the concentration–time curve, Cl (clearance), Vd (volume of distribution), Kel (elimination rate constant), and T_1/2_ (terminal half-life).

For the pharmacokinetic study in Beagle dogs, a total of 3 male and 3 female dogs were selected. They were acclimatized for 5 days before dosing and at the start of dosing, dogs were between 36 and 37 months old. Dogs were housed individually per sex in an environmentally controlled room, maintained at 24 ± 2 °C and 67 ± 1% relative humidity, on a 12–12 h light–dark cycle (beginning at 06:00 h) with 20 or 22 air changes per hour. Dogs were provided with 300 g feed every day and with water ad libitum. Dogs were observed twice daily for clinical, mortality, and morbidity signs. Sterile water was also selected as a vehicle for injection of compound **5**. Formulations were administered in dogs immediately after preparation by an intravenous bolus injection through the cephalic vein of the forelimb, using a sterile needle attached to a graduated syringe. A constant dose volume of 1 mL/kg body weight was administered, adjusted according to the most recently recorded body weight of each dog. A single dose of 5 mg/kg b. wt. was selected in the case of dogs.

Blood collection, plasma separation, and storage procedures were conducted following the same procedure as described for rats. Blood samples (600 µL) were collected from the cephalic vein of the forelimb for each dog. However, the concentration of compound **5** was determined using LC-MS/MS analysis. A column with identical characteristics to the one used in rats was employed, with a flow rate of 0.65 mL/min, a binary elution mode, and a run time of 6.0 min. Plasma samples were analyzed to determine the concentration of the active ingredient, and subsequently, the pharmacokinetic parameters were calculated. After a washout period of 7 days, dogs were returned back to stock.

#### 4.5.2. Maximum Tolerated Dose Study

A MTD study was performed for compound **5** in Wistar rats and Beagle dogs. A total of 25 male and 25 female Wistar rats were received in the experimental room and acclimatized for a period of 5 days prior to randomization. The environmental and housing conditions maintained during the experiment were identical to those described in the pharmacokinetic study. At the start of dosing, rats were between 6 and 7 w.o., and their body weight varied no more than ±20% from the mean body weight for each sex. During randomization, 20 male and 20 female rats were selected for the study, which were randomly allocated to four different dose level groups: D1, D2, D3, and D4. Each group consisted of 5 male and 5 female rats. The remaining 5 male and 5 female rats were returned to the animal facility. D3 and D4 rats were observed for a period of 14 days before being returned to the animal facility since MTD was achieved.

Dose formulations of compound **5** were administered by an intravenous bolus injection through the tail vein, using a sterile needle attached to a graduated syringe. Sterile water also served as a vehicle in the MTD assay. A fixed dose volume of 5 mL/kg b. wt. was used, and individual doses were adjusted according to the most recent recorded body weight of each rat. Dosing was performed with a gap of 48 h between each dose level. Rats were in observation for 14 days after dose administration. The dose levels used for rats were 300 mg/kg b. wt (D1) and 500 mg/kg b. wt. (D2).

Rats were subjected to twice-daily monitoring for clinical, mortality, and morbidity signs throughout the study. In addition, on the day of dosing, rats were observed at 1, 2, 3, 4 and 6 h after receiving the dose. Body weight was recorded on days 1 (before treatment), 4, 8, 11, and 14 of the study. At terminal sacrifice, rats were humanely euthanized using carbon dioxide asphyxiation, followed by a full gross necropsy. First, rats were examined for external abnormalities. Then, the cranial, thoracic, and abdominal cavities were dissected and a thorough examination of the organs was conducted. Tissue with gross lesions was discarded during report finalization.

In the case of dogs, one male and one female Beagle dog were selected for the study. A period of 5 days was used to allow dogs to acclimatize, and at the start of dosing, they were between 5 and 6 months old. Dogs were housed individually per sex in an environmentally controlled room, maintained at 24 ± 2 °C and 71.5 ± 4.5% relative humidity, on a 12–12 h light–dark cycle (beginning at 06:00 h) with 17 or 22 air changes per hour. Dogs were provided with 300 g feed every day and with water ad libitum.

Compound **5** was administered via an intravenous bolus injection through the cephalic vein of the forelimb, using a sterile needle attached to a graduated syringe. An initial dose level of 50 mg/kg b. wt. (D3 group) was selected based on MTD results for rats. Following a washout period of 7 days, a dose of 75 mg/kg b. wt. (D4 group) was administered to the same dogs based on observed toxicity. A constant volume of 1 mL/kg b. wt. was used, with individual doses adjusted to the most recent recorded body weight of each animal. Dogs were observed twice daily for clinical, mortality, and morbidity signs throughout the study period of 7 days. Body weight was recorded on dosing day and every two days during the assay period. Additionally, a daily record of individual dog feed intake and leftover feed was maintained. Blood samples were obtained prior to dosing and on day 7. Blood was extracted either from the cephalic vein in the forelimb or the lateral saphenous vein in the hind limb. Around 2.0 mL of blood was collected in non-heparin centrifuge tubes for serum separation to conduct clinical chemistry estimations. An additional 0.5 mL of blood was collected in EDTA-containing vials for hematology analysis and another 0.5 mL of blood was collected in tubes containing a 3.2% sodium citrate solution for Prothrombin time determination. Dogs were fasted overnight, with only water allowed prior to blood collection. Hematology and clinical chemistry determinations are available upon request. Dogs were euthanized by an intravenous injection of barbiturate (thiopentone sodium 100 mg/kg b. wt.). Then, a comprehensive gross necropsy was performed. Dogs were first examined for external abnormalities. The cranial, thoracic, and abdominal cavities were then dissected, and a thorough examination of the organs was conducted. Specifically, the liver, testes, uterus, brain, and kidneys were collected and subjected to microscopic examination using paraffin-embedded tissue stained with hematoxylin-eosin.

### 4.6. Statistical Analysis

All results are expressed as mean ± standard deviation (SD) and analyses were performed in triplicate. Statistical differences were analyzed using GraphPad Prism^®^ 9 (GraphPad Software, San Diego, CA, USA). Values of *p* < 0.05 were considered statistically significant and are indicated in graphical representations.

## 5. Conclusions

A thorough study of the mechanism of action of compounds **1** and **5** was presented in this work. Both molecules were proven to inhibit plasminogen’s lysine-binding sites without inhibiting the active sites of plasmin or tPA at therapeutic concentrations, or producing any effect on the coagulation pathway, showing similar behavior as that of TXA. Additionally, both molecules and TXA showed a similar antifibrinolytic profile. However, results demonstrated a higher antifibrinolytic activity at lower doses in whole blood for both compounds, and compound **1** even presented an enhancement of more than 2-fold over TXA in the D-dimer quantification. This difference can be attributed to their structural variations: the presence of a 1,2,3-triazole moiety extends the behavior of the molecules in the lysine binding sites, providing a more stable binding. These 1,2,3-triazole derivatives discovered by Bosch-Sanz et al. [43], and specially molecule **1**, present a promising opportunity for the development of novel antifibrinolytic drugs to overcome the limitations of the current gold standard.

## Figures and Tables

**Figure 1 ijms-25-07002-f001:**
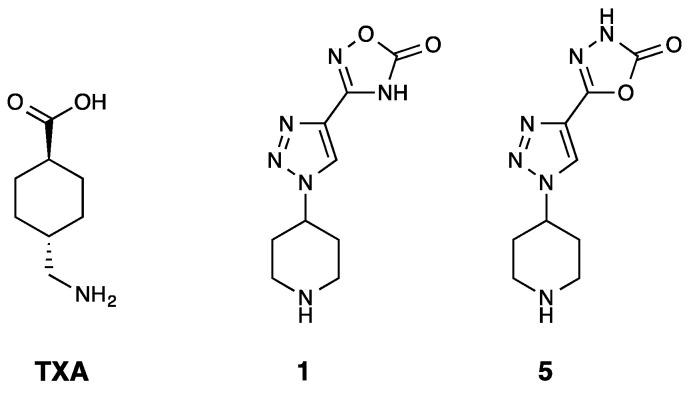
Structures of the current gold standard tranexamic acid (TXA) and the two 1,2,3-triazole derivatives discovered by Bosch-Sanz et al., which contained an oxadiazolone and a terminal piperidine ring [43]. The molecular weight of TXA is 157.21 g/mol, and for the hydrochloric salts of compounds **1** and **5**, the molecular weight is 272.69 g/mol.

**Figure 2 ijms-25-07002-f002:**
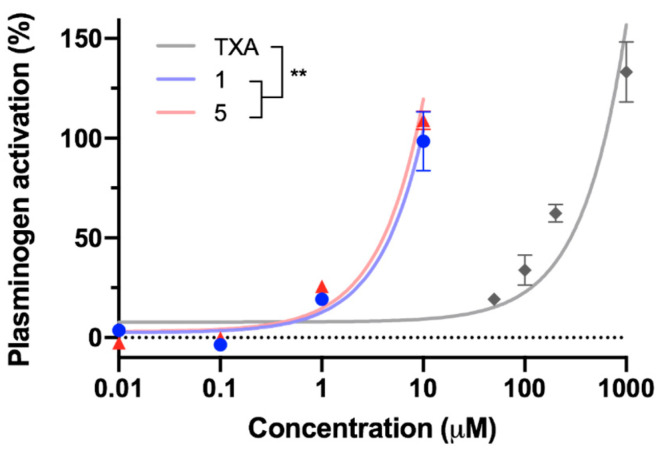
Activity of TXA and compounds **1** and **5** on the conversion of plasminogen into plasmin by tPA in the absence of fibrin. The percentage of plasminogen’s activation was determined by quantifying the increase in the enzyme’s activity relative to the activity in the absence of any compound. The concentrations at which each compound increased tPA’s activity by 50% were obtained for compound **1** (4.67 µM), compound **5** (4.03 µM), and TXA (283.01 µM). All datapoints are represented as mean ± SD. Data were fitted to a linear model and slopes were compared using an unpaired *t*-test with Welch’s correction. Statistically significant differences are indicated (** *p* < 0.01).

**Figure 3 ijms-25-07002-f003:**
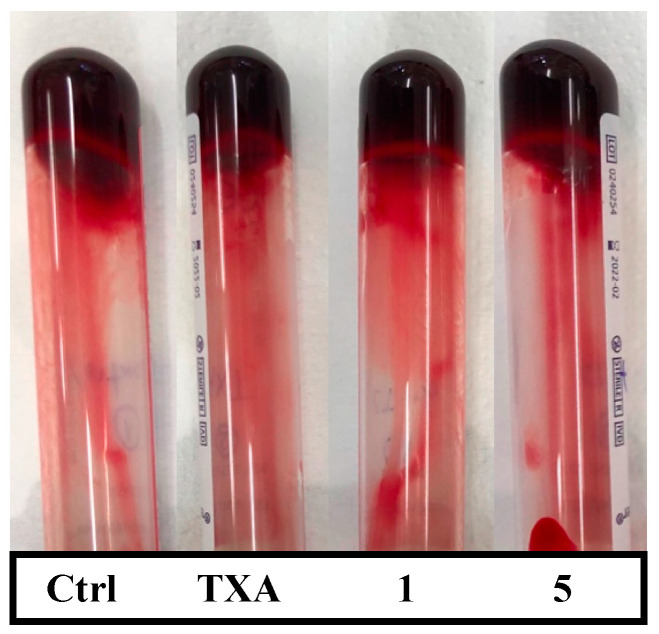
Blood clots containing the studied compounds after incubation at 37 °C for 30 min. Tubes were inverted to allow a better differentiation of clots (top) and liquid blood (bottom). In the control tube, no compound was added. TXA, **1**, and **5** tubes contained 40 µM of each product, respectively.

**Figure 4 ijms-25-07002-f004:**
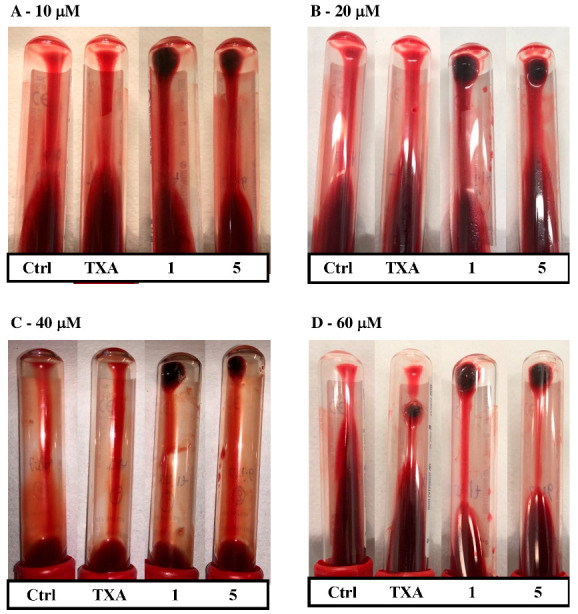
Images of the antifibrinolytic activity of compounds TXA, **1**, and **5** in whole blood clots. To initiate fibrinolysis, clots were treated with 10 µg/mL of tPA, and four different concentrations of each compound were tested: (**A**) 10 µM, (**B**) 20 µM, (**C**) 40 µM, and (**D**) 60 µM. Control tubes were clots to which only tPA was added. Tubes were inverted to allow a better differentiation of remaining clots (top) and liquid blood (bottom) after 24 h of incubation at 37 °C.

**Figure 5 ijms-25-07002-f005:**
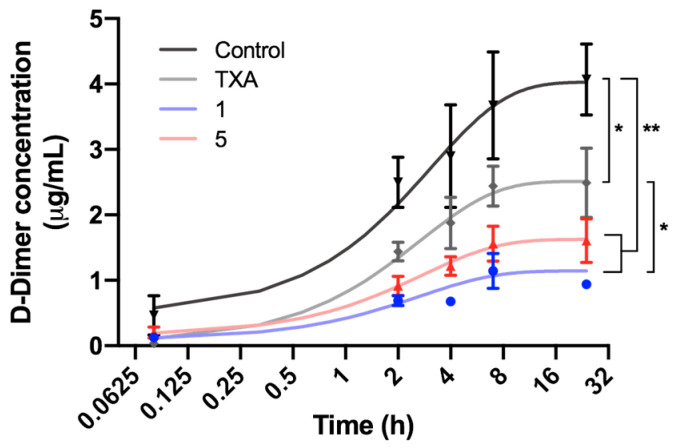
Effect of TXA and compounds **1** and **5** on fibrinolysis, quantified by D-dimer concentration during lysis of blood clots. Clots were incubated for 1 h before addition of exogenous tPA (final concentration 10 µg/mL) and 20 µM of TXA, compounds **1** or **5** (time 0 h). Control tubes did not contain any antifibrinolytic drug. Datapoints are presented as mean ± SD. Data were fitted to a nonlinear model and the slopes of the different conditions were obtained. Statistically significant differences, determined at timepoint 24 h using an unpaired *t*-test, are indicated (* *p* < 0.05, ** *p* < 0.01).

**Figure 6 ijms-25-07002-f006:**
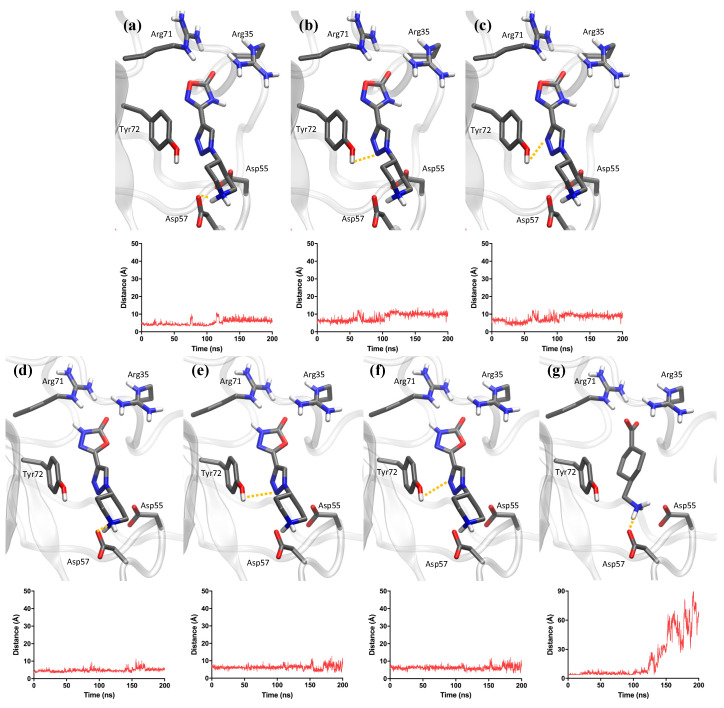
Representation of the most important distances for molecular dynamics simulations of compounds **1**, **5**, and TXA. Shown distances for compound **1**: (**a**) piperidine and Asp57, (**b**) triazole (nitrogen N2) and Tyr72, and (**c**) triazole (nitrogen N3) and Tyr72. Distances for compound **5**: (**d**) piperidine and Asp57, (**e**) triazole (nitrogen N2) and Tyr72, (**f**) triazole (nitrogen N3), and Tyr72. Distances for TXA: (**g**) terminal amine and Asp57.

**Figure 7 ijms-25-07002-f007:**
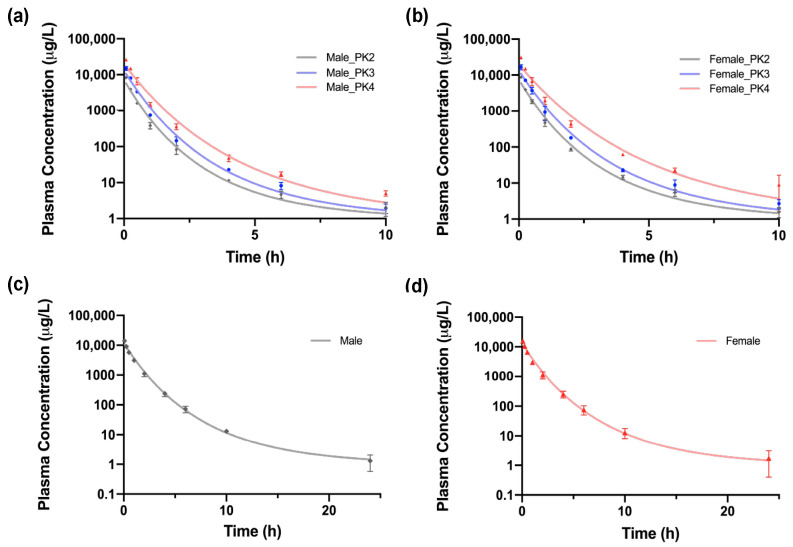
Plasma concentration in time curve of compound **5** after a single intravenous dose in (**a**) male and (**b**) female Wistar rats; and (**c**) male and (**d**) female Beagle dogs. Three different dose levels were studied for rats: 2.5 mg/kg (PK2), 5 mg/kg (PK3), and 10 mg/kg (PK4). For Beagle dogs, a single dose level of 5. mg/kg body weight (b. wt.) was used. All data points are the average of *n* = 3 animals per group. Datapoints are presented as mean ± SD. Data were fitted to a linear model.

**Table 1 ijms-25-07002-t001:** IC_50_ values for TXA and compounds **1** and **5** determined in the plasmin activity assay. IC_50_ was considered as the concentration at which each molecule inhibited plasmin’s cleavage of a synthetic substrate by 50%. IC_50_ values of compounds **1** and **5** were compared to TXA’s using an unpaired *t*-test and statistically significant differences are indicated.

Compound	IC_50_ (mM)	*p* Value
TXA	30.87 ± 5.50	
**1**	20.64 ± 0.78	0.0332
**5**	17.92 ± 1.24	0.0164

## Data Availability

The data presented in this study are available on request from the corresponding author. The data are not publicly available due to their length.

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
