# Peer review of "Unravelling the Antifibrinolytic Mechanism of Action of the 1,2,3-Triazole Derivatives"

_ijms, 2024, doi:10.3390/ijms25137002_

Round 1
Reviewer 1 Report
Comments and Suggestions for Authors
The authors presented an interesting scientific idea. However, studies conducted by molecular dynamics are subject to challenge, raising doubts about the results. From my perspective, the article is good and could be published with major revisions.
-The authors provided a set of bibliographic references in the introduction. However, the most recent reference provided by the authors is from 2020. The authors should conduct another literature review and add more recent articles to the introduction.
-It would be interesting for the authors to include the physicochemical properties of the molecules in the introduction.
-In the molecular dynamics section, the authors concluded about the molecules' permanence in the interaction sites. However, the simulation time is very short (18 ns) to conclude whether the molecule will remain at the site. The authors should extend the simulations to at least 100 ns to draw conclusions and perform simulation replicas to take an average.
-In their discussion, the authors referenced few articles reported in the literature to compare with their data. The authors should add more articles to their discussions.
-In the methodology, the authors mentioned using the PRODRG server to generate ligand topology files. However, PRODRG fails to generate partial charges for some bonds and chemical groups. Since the authors are planning to extend the simulation time to 100 ns, they could use another more reliable server like ATB (https://atb.uq.edu.au/), which is compatible with the GROMOS force field used.
Comments on the Quality of English LanguageMinor editing of English language required
Reviewer 2 Report
Comments and Suggestions for Authors
Dear authors,
I read the article very carefully and my recommendation is based on the observations made below.
"A set of compounds combining a piperidine ring, a triazole and an oxadiazolone ring were identified as potential antifibrinolytic drugs." - Please put a bibliographic index for this statement.
In my opinion, the justification for choosing the 2 compounds is unclear. They are the compounds synthesized by the authors. Have they already been tested as antifibrinolytics? The authors specify that they performed a docking. On what? And where are the references? I recommend the authors to rewrite part of the hypothesis in a clearer and better justified way. I also suggest renaming the compounds if the authors do not present them in a sequence of compounds. In the computational analysis part, I recommend the authors an explanation of the Kringle domain.
Why did the authors leave studying only the pharmacokinetic behavior of compound 5? Do the authors have approvals to conduct dog studies? And again in the discussion part, the authors present some results without references.
In my opinion, the study is conducted only on two substances whose choice is not convincingly justified. In addition, my ethical considerations do not justify conducting studies on dogs that are ultimately sacrificed.
My recommendation for this article is to reject it with the possibility of resubmission after completing the studies carried out on several compounds.
Round 2
Reviewer 1 Report
Comments and Suggestions for Authors
The authors have made the requested corrections. The article can be published in its current form.
Reviewer 2 Report
Comments and Suggestions for Authors
Dear authors,
thank you for taking into account the observations made. My recommendation is to publish as is.